# Chromosome-Level Genome Assembly Provides Insights into the Evolution of the Special Morphology and Behaviour of *Lepturacanthus savala*

**DOI:** 10.3390/genes14061268

**Published:** 2023-06-15

**Authors:** Ren-Xie Wu, Ben-Ben Miao, Fang-Yuan Han, Su-Fang Niu, Yan-Shan Liang, Zhen-Bang Liang, Qing-Hua Wang

**Affiliations:** College of Fisheries, Guangdong Ocean University, Zhanjiang 524088, China; benben.miao@outlook.com (B.-B.M.); hanfangyuan1999@163.com (F.-Y.H.); wolf0487@126.com (S.-F.N.); yanshan-liang@outlook.com (Y.-S.L.); liangzhenbang0403@163.com (Z.-B.L.);

**Keywords:** *Lepturacanthus savala*, genome sequencing, chromosomal assembly, comparative genomics

## Abstract

Savalani hairtail *Lepturacanthus savala* is a widely distributed fish along the Indo-Western Pacific coast, and contributes substantially to trichiurid fishery resources worldwide. In this study, the first chromosome-level genome assembly of *L. savala* was obtained by PacBio SMRT-Seq, Illumina HiSeq, and Hi-C technologies. The final assembled *L. savala* genome was 790.02 Mb with contig N50 and scaffold N50 values of 19.01 Mb and 32.77 Mb, respectively. The assembled sequences were anchored to 24 chromosomes by using Hi-C data. Combined with RNA sequencing data, 23,625 protein-coding genes were predicted, of which 96.0% were successfully annotated. In total, 67 gene family expansions and 93 gene family contractions were detected in the *L. savala* genome. Additionally, 1825 positively selected genes were identified. Based on a comparative genomic analysis, we screened a number of candidate genes associated with the specific morphology, behaviour-related immune system, and DNA repair mechanisms in *L. savala*. Our results preliminarily revealed mechanisms underlying the special morphological and behavioural characteristics of *L. savala* from a genomic perspective. Furthermore, this study provides valuable reference data for subsequent molecular ecology studies of *L. savala* and whole-genome analyses of other trichiurid fishes.

## 1. Introduction

Fish have the highest species diversity among vertebrates and highly diverse morphological and ecological properties [1]. More than 32,000 living fish species have been recorded to date [2]. Their size, morphology, physiological and behavioural characteristics, and adaptability vary greatly [3]. This variation has generated substantial interest in the development of genomic resources and assays of functionally important genes in fishes. With the development of genome sequencing and analytical methods, more and more genomic features of various fishes have been reported [4,5], such as zebrafish *Danio rerio* [6], tiger puffer *Takifugu rubripes* [7], yellowfin seabream *Acanthopagrus latus* [8], and giant grouper *Epinephelus lanceolatus* [9], etc. However, the substantial variation in biological properties and habitats is expected to correspond to significant differences in genome structure among different fishes [10]. Therefore, exploring the genomic evolution and adaptive mechanisms of various fishes has become a focus of animal genome research [11,12]. In particular, the genomes of wild fishes with special biological, behavioural, and ecological characteristics have gained widespread attention.

During long-term evolution, some fishes have undergone substantial divergence in morphology, habits, behavioural traits, and survival and propagation strategies [13,14]. These traits often involve complex evolution of the genome and related developmental mechanisms. For example, cave fish commonly exhibit a series of specific phenotypic changes, including eye degeneration, pigment loss, and increases in taste buds and mechanosensory organs [15,16]. McGaugh et al. [17] identified candidate genes that cause eye degeneration by sequencing the genome of the Mexican tetra *Astyanax mexicanus*. A genomic analysis of the elephant shark *Callorhinchus milii* showed that the *SCPP* (secreted calcium-binding phosphoprotein) gene loss explains the absence of hard bones in the endoskeleton of cartilaginous fishes [18]. In the tiger tail seahorse *Hippocampus comes*, the disappearance of its ventral fins may be related to the *TBX4* gene loss, which regulates hindlimb formation [19]. Moreover, the high expression of the expanded astacin metalloprotease gene family in the brood pouch of male *H. comes* contributed to its pregnancy [19]. Based on comparative genomic analyses, the accelerated evolution of genes involved in the growth hormone and insulin-like growth factor 1 axis was revealed to be an important driving factor for the rapid growth and large size of ocean sunfish *Mola mola* [20]. In comparative genomic analyses of Siamese fighting fish *Betta splendens* and its five variants, a large number of single nucleotide polymorphisms (SNPs) and genes related to aggressive behaviour have been detected [21]. Recently, Zhao et al. [22] reported that fast-swimming fishes (e.g., southern bluefin tuna *Thunnus maccoyii*, Pacific bluefin tuna *Thunnus orientalis*, swordfish *Xiphias gladius*, and large yellow croaker *Larimichthys crocea*) have more haemoglobin genes than relatively slow-moving fishes (e.g., ocean sunfish *Mola mola*, tongue sole *Cynoglossus semilaevis*, and *H. comes*). These research advances provide insights into the formation of unique phenotypic and behavioural traits in wild fishes at the genomic level.

The Savalani hairtail *Lepturacanthus savala* (Cuvier, 1829), which belongs to the family Trichiuridae (Teleostei, Perciformes), is a benthopelagic fish widely distributed in the tropical and subtropical waters of the Indo-West Pacific region [23,24]. It is one of the main fishing targets for bottom trawls, shore seines, and bag nets in the coastal countries of Asia [25]. In China, *L. savala* can be found in the East China and South China Seas, and is abundant in the northern South China Sea [26]. Based on a routine fishery resources survey, the annual catch of *L. savala* was one-quarter to one-fifth of the total annual catch of trichiurids in the northern South China Sea (approximately 300,000 t, 2010–2021), supporting an important commercial marine fishery. In some Indian Ocean countries, *L. savala* is also a major fishery resource [27]. In the period of 1999 to 2009, the annual catch of this species in Pakistani coastal waters ranged from 31,623 t to 20,375 t [28]. *L. savala* is a popular hairtail fish and contributes substantially to the world trichiurid fisheries, second only to the genus *Trichiurus*.

Similar to other trichiurid fishes, the body of *L. savala* is remarkably elongated and strongly compressed, with a ribbon-like shape. Its total length generally ranges from 30 to 87 cm (maximum about 100 cm) [29], and the number of vertebrae (135 to 141) [30] exceeds that of most teleost fishes (mainly 21 to 56) [31]. Both the ventral and caudal fins of *L. savala* are absent, with a whip-elongated tail that is grey-black at the end [32]. The first anal-fin spine is large, its length half of the diameter of the eye, and the two small canine teeth on the upper jaw project forward [25]. These taxonomic traits distinguish *L. savala* from other trichiurid species. As a ferocious predatory fish, *L. savala* occupies a high trophic level in the marine food chain [33]. It not only has extremely sharp teeth, but also has strong swimming ability [26]. Moreover, the long-distance migratory behaviour of *L. savala* [34] is supported by its excellent swimming motility [35]. *L. savala* is relatively derived and is in a special evolutionary position in the family Trichiuridae. *L. savala* is also clearly distinguished from other teleost fishes and can be considered a special case in the genetic evolution of teleosts. Previous studies of *L. savala* have mainly focused on fishery resources [28,36], biological characteristics [32], feeding habits [37], mitochondrial DNA [38], and population genetics [39]. However, the molecular mechanisms underlying the evolution of its unique traits have not yet been addressed. Therefore, genomic studies of *L. savala* are needed to elucidate the evolutionary mechanisms underlying its particular morphological and behavioural traits, and also to unravel the molecular determinants of the formation of this special group of trichiurid fishes.

In this study, we combined Illumina short reads, PacBio long reads, and Hi-C sequencing data to obtain a chromosome-level genome assembly of *L. savala*. RNA sequencing of muscle, liver, and heart tissues was performed using PacBio and Illumina platforms to assist in the structural and functional annotations of the genome. Finally, we performed the comparative analyses and searched for the signature of positive selection in genomic data for *L. savala* and other fish species to investigate phylogenetic relationships, divergence times, and gene family contraction and expansion. Our findings clarify the evolutionary mechanisms underlying specific features of *L. savala* at the genome-wide level, and provide important genomic resources and new perspectives for exploring the genomic evolution of trichiurid fishes. 

## 2. Materials and Methods

### 2.1. Sample Collection

During the fishery resource survey along the eastern coast of Leizhou Peninsula (Zhanjiang City, Guangdong Province, China) in December 2020, three male *L. savala* were captured through bottom trawls (Figure 1). Live fish were anaesthetized using MS-222 (ethyl 3-aminobenzoate methanesulfonate, Sigma-Aldrich, Shanghai, China) at a concentration of 200 mg/L. After the fish were deeply anaesthetized, the muscle, liver, and heart tissue samples were collected from each fish using three 1.5 mL sterile tubes. To avoid contamination, the sampled tissues were not mixed with other tissues (e.g., gills, intestines, and stomach) or environmental DNA. The samples were immediately placed in liquid nitrogen for rapid freezing, and then transferred to a −80 °C refrigerator in the laboratory for subsequent construction of sequencing libraries. The experimental animal protocols in this study were reviewed and approved by the Animal Experimental Ethics Committee of Guangdong Ocean University, China (approval number: 1201-2020).

### 2.2. DNA and RNA Extraction for Library Construction and Sequencing

The genomic DNA was extracted from muscle tissue using the standard phenol/chloroform extraction protocol [40]. The concentration of extracted DNA was detected by Nanodrop 2000 Spectrophotometer (Thermo Fisher Scientific, Waltham, MA, USA), and the purity and integrity were determined by agarose gel electrophoresis. According to the standard Illumina protocol, a paired-end library with an insert size of 150 bp was constructed for sequencing on the Illumina HiSeq 2500 platform (Illumina, San Diego, CA, USA). A SMRTbell library with a fragment size of 20 kb was constructed using the SMRTbell Express Template Prep Kit for sequencing on the PacBio Sequel II platform (Pacific Biosciences, Menlo Park, CA, USA). DNA samples were fragmented by Covaris M220 ultrasonic disruptor (Covaris, Shanghai, China), followed by enrichment and purification of large DNA fragments using magnetic beads. 

Total RNA was extracted using TRIzol reagent (Invitrogen, Carlsbad, CA, USA) for Illumina library construction for each tissue type (the muscle, liver, and heart), while the PacBio library was constructed using mixed RNA from the three tissues. The RNA integrity number (RIN) and RNA concentration of each extracted total RNA were then detected by Agilent 2100 Bioanalyzer (Agilent Technologies, Palo Alto, CA, USA) and agarose gel electrophoresis, respectively. Total RNA of each tissue was reverse transcribed using the TUREscript First Stand cDNA Synthesis Kit (AidLab, Beijing, China), and a double-stranded cDNA library was synthesized, resulting in the construction of three Illumina paired-end sequencing libraries with insert sizes of about 150 bp. Full-length cDNA was synthesized using the SMARTer cDNA Synthesis Kit (Takara Bio, Beijing, China), and the cDNA concentration in the library was measured using Qubit 3.0 Fluorometer (Invitrogen, Carlsbad, CA, USA). Subsequently, the full-length cDNA fragments were end-repaired, and the SMRT dumbbell adapters were connected for constructing the PacBio sequence library. Finally, the libraries were sequenced using the Illumina HiSeq 2500 and PacBio Sequel II platforms, and the resulting data were used for genome annotations.

### 2.3. Evaluation of Genome Size, Heterozygosity, and Contamination

The raw reads obtained by Illumina sequencing of muscle DNA were filtered using Trimmomatic v0.39 [41] to obtain clean reads. Ten thousand randomly selected clean reads (5000 for Read1 and 5000 for Read2) were mapped to the NCBI nucleotide database by using NCBI blast++, and the top six mapped species were selected in descending order of mapping time. If all mapped results are homologous, the samples have not been subjected to exogenous contamination. Jellyfish v1.1.11 [42] was used to mathematically estimate the genome size based on the K-mer analysis method (base sequence containing K bases). The 17 bp K-mers (17mers) were extracted from the sequencing data, and the frequency of each 17mer was calculated. The K-mer depth was the expected value corresponding to the Poisson distribution. The calculated genome size (unit: Megabits) was defined as the number of K-mers/depth of K-mers. Total K-mers were assembled using SOAPdenovo v2.0.0 [43] with K-mer set at 41 bp. The heterozygosity rate of the corrected genome was obtained by calculating the proportion of heterozygous sites.

### 2.4. Genome Assembly and Integrity Assessment

The raw sequencing data contained two adaptors, which was the dumbbell-shaped structural sequence called polymerase reads. Subreads were obtained after the adaptor sequences were interrupted and filtered out, then the high-precision HiFi reads were generated using SMRT Link v10.2 [44]. Hifiasm v0.12 [45] was used to quickly assemble HiFi reads (parameters selected by default), and the contigs and scaffolds with more complete sequences were constructed in turn. After obtaining contigs, NextPolish v1.2.1 [46] was used to correct errors with Illumina clean reads obtained in the genome survey step, and finally obtain more accurate genome sequences.

Integrities of the assembled genome and the conserved genes in the assembled genome were judged using Benchmarking Universal Single-copy Orthologs method (BUSCO) [46] and Core Eukaryotic Genes Mapping Approach (CEGMA) [47], respectively, to evaluate the integrity of the assembled genome and the uniformity of Illumina and PacBio sequencing. Illumina clean reads were mapped to the assembled genome sequences using BWA v0.7.17 [48]. Further, the BWA mapping results (BAM format) were used to detect SNPs at the genome scale using samtools v1.15.1 [49]. The processing included sorting chromosome coordinates and removing duplicate reads, etc. The SNPs contain homology SNPs and heterozygosis SNPs, and the ratio of homology SNPs to total SNPs could reflect the correctness of the genome assembly.

### 2.5. Chromosome Assembly by Hi-C

Hi-C libraries of muscle tissue were built according to the high-throughput chromatin conformation capture (Hi-C) library construction technology standards, and then sequenced using the Illumina HiSeq 2500 platform. Only filtered reads that passed the HiCUP v0.8.0 [50] quality control pipeline were used for subsequent chromosome assembly. HiCUP subroutine hicup_truncater was used to identify the restriction sites on clean reads and cut off redundant chimeric sequences. Paired-end reads were mapped to the preliminarily assembled genome using the hicup_mapper subroutine, and the mapping results were combined. The resulting data were filtered using the hicup_filter subroutine to obtain valid pairs used as Di-Tags. Thereafter, PCR repeats were removed by the hicup_deduplicater subroutine. The data obtained after quality control contained effective genome-wide chromosome cross-linking information, which facilitated genome assembly to the chromosome level. Since the interaction frequency on the same chromosome decreases as the interaction distance increases, the contigs or scaffolds of the same chromosome can be sorted and oriented. Accordingly, ALLHiC program [51] was used to assemble the Hi-C data and to cluster the assembled contig/scaffold sequences to obtain a chromosome-level genome.

### 2.6. Genome Repetition, Structure, Function, and Noncoding RNA Annotation

Protein-coding genes were predicted by using three methods, ab initio prediction, homology-based identification, and an RNA-Seq data-assisted method. First, Augustus v3.4.0 [52], GlimmerHMM v3.01 [53], Geneid v1.4.4 [54], and Genscan v1.0.0 [55] were used for ab initio prediction by counting codon frequency, exon–intron distribution, and training dataset, etc. The RNA-Seq data from three tissues act as the input training sets of Augustus and SNAP programs. Second, genome-wide protein sequences of *D. rerio*, *Homo sapiens*, *T. maccoyii*, *Thunnus albacares*, *Etheostoma spectabile*, *Sander lucioperca*, *Perca fluviatilis*, *Perca flavescens*, *T. rubripes*, *Gasterosteus aculeatus*, and *Oryzias latipes* were downloaded from the NCBI database and used for homology mapping to the *L. savala* genome using tBlastn [56] and GeneWise v2.4.1 [57] in order to identify known genes with high similarity (E-value ≤ 1 × 10^−5^). Third, two methods were used to predict protein coding genes based on RNA-Seq data from three tissues. That is, the prediction gene after Tophat v2.1.1 [58] and Trinity v2.13.2 [59] assembly were performed using Cufflinks v2.2.1 [58] and PASA v2.5.2 (https://github.com/PASApipeline/PASApipeline, accessed on 21 March 2022), respectively. Gene sets predicted by the above three methods were integrated by EVidenceModeler v1.1.1 [60]. Alternative splicing transcripts were removed, and the longest transcripts were retained using PASA. Further, the predicted gene sequences were mapped to NR (nonredundant proteins), SwissProt (Swiss Protein Institute), KEGG (Kyoto Encyclopedia of Genes and Genomes) [61], and GO (gene ontology) [62] databases for the functional annotation of protein-coding genes. Conserved functional domain information and protein families were predicted using the Pfam (protein family) [63] and InterPro (Integrated Resource of Protein) databases.

Tandem repeats in the genome were searched using TRF program. Database mapping and ab initio prediction methods were used to identify interspersed repeats in the genome. Based on the homologous repeat database RepBase [64], RepeatMasker v4.1.2, and RepeatProteinMask v4.1.2 [65], programs were used to identify sequences with similar repeat sequences of known nucleic acids and amino acids, respectively. The ab initio prediction method firstly used LTR_Finder v1.0.7 [66], RepeatScout v1.0.5 [53], and RepeatModeler v2 [53] programs to build de novo repeated sequence database, and then RepeatMasker v4.1.2 was used to predict interspersed repeats. Annotations of noncoding RNA included tRNA, rRNA, miRNA, and snRNA. tRNAscan-SE [67] was used to predict tRNA. The rRNA sequences of closely related species were selected as reference sequences for searches by blast alignment. MiRNAs and snRNAs were predicted based on the Rfam family covariance model using INFERNAL.

### 2.7. Genome Evolution, Gene Family Dynamics, and Positive Selection Analyses

Protein-coding genes of less than 30 amino acids in the genomes of *L. savala* and 18 other fishes (*Acanthopagrus schlegelii*, *C. semilaevis*, *D. rerio*, *Epinephelus akaara*, *G. aculeatus*, *H. comes*, *Ictalurus punctatus*, *L. crocea*, *Monopterus albus*, *P. flavescens*, *Seriola dumerili*, *Scleropages formosus*, *Scophthalmus maximus*, *Sebastes schlegelii*, *T. albacares*, *T. maccoyii*, *T. rubripes*, and *Cetorhinus maximus*) were removed, and alternative splicing of the longest transcript was used for a gene family cluster analysis. Similarity relationships between the protein sequences of these 19 species were calculated by Blastp [56] (E-value = 1 × 10^−7^). Based on similarity, orthologous genes from 19 species were clustered using OrthoMCL v2.0.9 [68] (extension coefficient = 1.5) to obtain single-copy gene families and multi-copy gene families. Then, the expansion and contraction of gene families were evaluated using CAFE (http://sourceforge.net/projects/cafehahnlab/, accessed on 25 March 2022). Further, GO and KEGG pathway enrichment analyses of expanded and contracted gene families were carried out using BioSciTools (https://bioscitools.github.io, accessed on 25 March 2022) with *p* < 0.05 and FDR (false discovery rate) < 0.05 as thresholds for statistical significance. Gene families detected only in *L. savala* and not in other species were considered to be unique to *L. savala*.

A maximum likelihood phylogenetic tree was constructed using the RAxML program [64] based on alignment of all single-copy genes for the 19 species. Divergence times between species (95% confidence intervals) were estimated using McMcTree in the PAML v1.3.1 package [69]. Six calibration times for species divergence were obtained from the TimeTree database [70], including *G. aculeatus* and *S. schlegelii* (68–87 Mya), *C. semilaevis* and *S. maximus* (49–81 Mya), *T. rubripes* and *G. aculeatus* (99–127 Mya), *T. rubripes* and *C. semilaevis* (94–115 Mya), *T. rubripes* and *H. comes* (106–114 Mya), and *T. rubripes* and *D. rerio* (206–252 Mya). Finally, the convergence of bifurcation times for tree branches was verified by Tracer v1.7.1 [71].

Candidate genes associated with the special traits of *L. savala* were screened by setting up two groups of positive selection combinations, and gene function annotation and enrichment analyses were performed. Group 1 (*L. savala*) vs. a selection of fishes (*A. schlegelii*, *L. crocea*, and *P. flavescens*) was used to screen the genes associated with the ribbon-like shape, scaleless body surface, and absence of ventral fins of *L. savala*. Group 2 (*L. savala*, *M. albus*) vs. the same selection of fishes (*A. schlegelii*, *L. crocea*, and *P. flavescens*) was set to further screen for genes associated with the body shape of *L. savala*. Based on protein sequence alignment data for single-copy gene families of the above five species, the branch site model in PAML was used to detect whether each gene family was positively selected in the foreground branch. Finally, GO and KEGG enrichment analyses of positively selected genes were performed using BioSciTools, and *p* < 0.05 and FDR < 0.05 were used as thresholds for statistical significance.

## 3. Results

### 3.1. Genome Size Estimation and Initial Characterization of the Genome

In total, 291,902,400 raw paired-end reads were generated by genomic surveys on the Illumina platform, and 237,372,083 clean reads were obtained for subsequent analyses. The proportion of clean reads with base quality > Q30 was 90.66%, the sequencing error rate was 0.04%, and the GC content was 39.75%. Mapping results showed that the top six species were all Perciformes, namely, *Dicentrarchus labrax* (0.36%), *Haplochromis burtoni* (0.32%), *Tetraodon nigroviridis* (0.2%), *T. rubripes* (0.19%), *O*. *latipes* (0.11%), and *Trichiurus lepturus* (0.11%). This indicates that the sequencing data were reliable and free from genomic contamination by other species, especially microorganisms. Based on the expected value of the Poisson distribution given by the K-mer analysis (K-mer = 17), the K-mer depth was 78. Accordingly, the genome size of *L. savala* was estimated to be 815.49 Mbp, revised to 802.34 Mbp, and the genome heterozygosity rate was 0.53%.

### 3.2. Genome Assembly and Evaluation

A total of 1,591,638 high-quality HiFi reads were obtained by PacBio-SMRT sequencing, and 215 contigs were assembled (Appendix A). The assembled genome size was 790.02 Mbp, close to the estimate from the genome survey (802.34 Mbp), and the contig N50 length reached 19.01 Mbp. 

Taking the database with 3640 orthologous single-copy genes constructed by BUSCO as a reference, the *L. savala* genome contained 3491 (95.9%) complete BUSCOs, of which 3459 (95.0%) were complete single-copy BUSCOs, and 32 (0.90%) were complete duplicated BUSCOs (Appendix A). That is, the assembled genome contained more than 95.9% of orthologous genes, indicating a high rate of gene coverage. A CEGMA evaluation showed that the assembled genome completely matched 229 (92.34%) of 248 conserved genes in eukaryotic model organisms, and that the conserved genes were fully assembled, demonstrating the integrity of the *L. savala* genome. The read mapping rate was as high as 98.16%, the proportion of the genome covered by reads was 99.91%, and the average depth of coverage per base by reads was 86.79%. Furthermore, 2,498,432 (0.3918%) heterozygous SNPs and 641 (0.0001%) homologous SNPs were identified. The low ratio of homologous SNPs indicated a high single-base accuracy for the assembled genome. The GC content of the assembled genome sequences calculated with a 10 Kbp window was concentrated around 39.03%, and no significant GC separation was detected, indicating a lack of exogenous contamination in the *L. savala* genome.

### 3.3. Chromosome Assembly by Hi-C Data

A total of 12,034,266 raw paired-end reads were generated by the Illumina sequencing of the Hi-C library, and 10,213,512 clean reads were obtained after quality control. A HiCUP analysis and mapping results showed 6,124,460 clean reads (59.96%) for read1 and 6,124,460 (59.96%) clean reads for read2. There were 5,365,963 valid Di-Tags and 758,497 invalid Di-tags (containing multiple types) obtained by hup_filter filtering (Appendix A). After removing PCR repeats, 5,117,719 unique Di-Tags were retained and 2,189,442 unique cis Di-Tags (560,836 cis-close Di-Tags and 1,628,606 cis-far Di-Tags) and 2,928,442 unique trans Di-Tags were identified. Thus, the effective utilization of Hi-C data, calculated as unique Di-Tags/total read pairs, was 50.11%. These Di-Tags record the frequency of interactions within and among chromosomes, and the assembled chromosome-level genome contained 219 contigs and 101 scaffolds. Chromosome clustering based on 11 scaffolds (790,034,746 bp) showed that 24 sequences (758,527,366 bp) were anchored and 77 sequences (31,507,380 bp) were not anchored to the chromosome, with a genome assembly rate of 96.01% (Figure 2).

### 3.4. Genome Annotation

After quality control and filtering, Illumina sequencing of RNAs from muscle, liver, and heart tissues yielded 22,926,095, 18,722,468, and 20,707,056 clean reads, with Q30 values of 93.77%, 93.64%, and 93.38%, respectively, and GC contents ranging from 49.12% to 50.89% (Figure 3). The PacBio SMRT sequencing of mixed RNA from three tissues generated 529,509 polymerase reads (46.52 Gbp) with an average length of 87,857 bp and an N50 length of 159,345 bp, as well as 13,516,264 subreads (45.51 Gbp) with an average length of 3368 bp and an N50 length of 3721 bp. These transcriptomic data were used to assist in genome annotation.

A total of 31,876 genes were predicted by three methods (Table 1, Appendix A). The longest transcript was selected by filtering out alternative splicing variants to obtain 23,625 protein-coding genes in the *L. savala* genome (Table 1). Basic information for 22,670 and 20,571 genes was obtained from the NR and SwissProt databases, respectively. Biological processes and functions for 15,555 and 20,399 genes were derived from the GO and KEGG databases, respectively. Annotation information for functional domains and protein families for 18,926 and 20,429 genes were acquired from the Pfam and InterPro databases, respectively. Integrating these results, 22,679 (96.0%) genes were successfully annotated, of which 18,955 genes had complete annotation information in the above six databases (Appendix A).

In total, 96,078,789 bp of tandem repeats were predicted using TRF, accounting for approximately 12.16% of the genome. A total of 277,525,905 bp of interspersed repeats were identified by integrating the results of database mapping and ab initio prediction, accounting for approximately 35.13% of the genome (Appendix A). Annotation of nc-RNAs showed that the *L. savala* genome had 1434 miRNAs (143,870 bp; 0.0182%), 9086 tRNAs (685,943 bp; 0.0868%), and 10,263 rRNAs (2,100,089 bp; 0.27%) (Appendix A).

### 3.5. Gene Family Clustering, Expansion and Contraction, and Phylogenetic Analyses

The protein-coding genes screened from the genomes of *L. savala* and 18 other fishes were in the range of 18,785 (*A. schlegelii*) to 25,573 (*D. rerio*), and a cluster analysis generated 20,932 genes, of which 2,068 were single-copy gene families (Figure 4A). There were 13,907 gene families shared by *L. savala* and three closely species (*T. albacares*, *T. maccoyii*, and *P. flavescens*), and 407 gene families were unique to *L. savala* (Figure 4B). KEGG enrichment analysis showed that these unique gene families were mainly involved in the following pathways: protein digestion and absorption, PI3K-Akt signaling pathway, focal adhesion, ECM–receptor interaction, platelet activation, relaxation signaling pathway, lysine degradation, and amoebiasis. 

Genome family expansion and contraction were further analyzed for 20,932 gene families in 19 species. Based on a comparison with the common ancestors of *L. savala*, *T. albacares*, and *T*. *maccoyii*, 67 gene families expanded and 93 gene families contracted during the evolution of *L. savala* (Figure 5). KEGG enrichment analysis (Table 2) revealed that the expanded gene families were involved in several important pathways, such as focal adhesion, ECM–receptor interaction, platelet activation, relaxation signaling pathway, protein digestion and absorption, PI3K-Akt signaling pathway, lysine degradation, cortisol synthesis and secretion, and PPAR signaling pathway. These pathways were highly consistent with the pathways associated with the unique gene families of *L. savala*. The main pathways related to the contracted gene families included synaptic vesicle cycle, GABAergic synapse, NOD-like receptor signaling pathway, protein digestion and absorption, mineral absorption, arachidonic acid metabolism, ECM–receptor interaction, and focal adhesion (Table 2). 

As illustrated in the phylogenetic trees (Figure 5 and Figure 6), *L. savala*, *T. albacares* (BioProject: PRJEB47267), and *T. maccoyii* (BioProject: PRJEB46021) were first clustered into a monophyletic clade with 100% bootstrap support, and all nodes of other branches also showed 100% support. As shown in Figure 6, the divergence between *L. savala* with *T. maccoyii* and *T. albacares* occurred 84.4 (60.1–107.6) million years ago, while *T. maccoyii* and *T. albacares* diverged 3.6 (2.9–4.4) million years ago. 

### 3.6. Positive Selection Analysis

A total of 903 genes were identified in the first positive selection analysis (Table 3). These genes were mainly enriched in the GO terms with DNA metabolic process, nuclear chromosome, and DNA repair, and in the KEGG pathways with JAK-STAT signaling pathway, novobiocin biosynthesis, Fanconi anemia pathway, cytokine–cytokine receptor interaction, homologous recombination, nonhomologous end-joining, and complement and coagulation cascades (Appendix A). In the second positive selection analysis, 922 genes were identified (Table 3). The genes were mainly enriched in the GO terms with methyltransferase activity, amino methyltransferase activity, and nucleic acid binding, and in the KEGG pathways with cytokine–cytokine receptor interaction, JAK-STAT signaling pathway, RNA transport, autophagy-other, autophagy-yeast, Fanconi anemia pathway, and nonhomologous end-joining (Appendix A). 

We further evaluated the correlations between the functions of positively selected genes and gene families in *L. savala* and its biological characteristics, and finally confirmed that gene families *TES*, *TRIO*, *DNAH*, *SLC6*, and *COL4*, the genes *MTOR*, *ATG3*, *ATG4C*, *ATG12*, *CFI*, *C1QA*, *VTN*, *STAT6*, *IL5RA*, *IL10*, *IL15RA*, *IL16*, *IL17RA*, *IL20RA*, *IL22RA2*, *POLM*, *PRKDC*, *BARD1*, *BRCA1*, *NBN*, *XRCC2*, *EME2*, and *FAAP100*, and autophagy-other, complement and coagulation cascades, JAK-STAT signaling pathway, cytokine–cytokine receptor interaction, nonhomologous end-joining, homologous recombination, Fanconi anemia pathways play important roles in the evolutionary of unique traits in *L. savala*. 

## 4. Discussion

### 4.1. Quality Evaluation of the L. savala Genome

We obtained the first high-quality genome assembly at the chromosome-level of *L. savala* by combining PacBio SMRT-Seq, Illumina HiSeq, and Hi-C technologies. The Q20 and Q30 scores of raw data were all greater than 90%, indicating that the sequencing data were of high quality and could be used for subsequent analyses. The genome size and GC content of *L. savala* were 790.02 Mbp and 39.03%, respectively, which were roughly equivalent to those of *T. albacares* (792.10 Mbp, 39.5%) and *T. maccoyii* (782.42 Mbp, 39.5%,). Based on a genome survey, Song et al. [72] reported that the genome sizes of *Trichiurus japonicus*, *Trichiurus nanhaiensis*, *Trichiurus brevis*, *L. savala*, and *Eupleurogrammus muticus* from the coastal waters of China were 913 Mb, 868 Mb, 871 Mb, 747 Mb, and 670 Mb, respectively, with average GC contents of 39.59% to 42.05% and repeat sequence contents of 33.21% to 45.87%. Our data were consistent with these previous estimates. As expected, the final number of chromosomes assembled was 24 for *L. savala*, as well as for *T. albacares* and *T. maccoyii*. A phylogenetic tree supported the relatively close relationships among these three species, consistent with morphological classification results. 

Heterozygosity reflects the difficulty of whole-genome sequencing and assembly [73]. The genomic heterozygosity rate of 0.53% in our study was slightly lower than that (0.72%) reported by Song et al. [72]. This may be related to the different K-mer depths obtained by the two survey analyses (78 in our study and 45 in the previous study). According to the repeat sequence content (40.54%) and heterozygosity rate (0.53%) of *L. savala*, we believed that the *L. savala* genome was a typical diploid genome. Additionally, contig N50 and scaffold N50 values are important indexes for judging the quality of species genomes [74]. The contig N50 and scaffold N50 obtained for the assembly of the *L. savala* genome were 19,013,249 bp and 32,774,443 bp, respectively, which were similar to other fishes reported in recent years [75,76,77]. Such high-quality genomic data provide a reliable basis for studies of the special morphological and behavioural characteristics of *L*. *savala* at the genomic level. 

### 4.2. Genes Associated with the Specific Morphology of L. savala

In this study, we detected a significant expansion of the *TES* gene family in *L. savala*. *TES* encodes a novel focal adhesion protein that contains three C-terminal LIM domains, and is involved in cell motility and adhesion [78]. This protein is widely expressed in normal tissues of animals, and may play key role in the reorganization of the actin cytoskeleton [79,80]. Dingwell and Smith [81] demonstrated that TES protein deficiency caused a sharp decrease in the number of posterior trunk and tail somites during embryonic development in the African clawed frog *Xenopus laevis*. This indicated that the *TES* gene plays a crucial role in regulating axial elongation in *X. laevis* in the late gastrula stage. The expansion of the *TES* gene family is likely to be essential for the formation of the elongated ribbon body axis in *L. savala*. In positive selection analysis (group 2), we screened several key genes that were significantly enriched in the autophagy pathway, such as *MTOR*, *ATG3*, *ATG4C*, and *ATG12*. *MTOR* encodes phosphatidylinositol kinase-related kinases, composed of two complexes (mTORC1 and mTORC2) [82,83]. mTORC1 controls protein synthesis, cell growth, and proliferation [84]. As a pivotal regulator of skeletal growth [85], mTORC1 plays an important role in the growth of long bones in mice by regulating the proliferation and differentiation of chondrocytes [86]. mTORC2 is a regulator of the actin cytoskeleton and promotes cell survival and cell cycle progression [87]. Chen et al. [88] reported that mTORC2 signaling mediated by Rictor (a core subunit of mTORC2) plays a crucial role in promoting chondrocyte hypertrophy and enhancing osteoblast activity in mice. Accordingly, we infer that the *MTOR* gene promotes the proliferation and differentiation of *L. savala* vertebrae, resulting in a significantly greater number of vertebrae than is found in most teleost fishes.

Moreover, the autophagy-related proteins encoded by the *ATG* gene family screened here are essential for autophagosome formation [89], and play pivotal roles in the autophagic process [90,91]. *ATG3* encodes a ubiquitin-like-conjugating enzyme, which is a component of the autophagy-related ubiquitination-like systems, and is involved in autophagosome formation [92,93]. *ATG4C* encodes a cysteine protease that plays an essential role in autophagy by mediating both proteolytic activation and delipidation of ATG8 family proteins [94,95]. Autophagy is an important pathway in many developmental processes in higher eukaryotes [96]. It is involved in apoptosis and tissue remodeling during embryogenesis [97], and is responsible for the degradation of normal proteins during animal metamorphosis and development [96]. Autophagy is also induced by amino acid deficiencies in the animal starvation response [98,99]. Previous studies have demonstrated that the remodeling of larval organs in most lepidopterans during metamorphosis involved autophagy, which is considered essential in the process of organ degeneration in arthropods [100,101]. Franzetti et al. [102] observed that the expression levels of autophagy-related genes (*ATG5*, *ATG6*, and *ATG8*) in the midgut cells increase significantly during midgut remodeling of the larvae of the silkworm *Bombyx mori*. Autophagy is a prerequisite for the regeneration of the caudal fin in *D. rerio*, which promotes the survival and differentiation of blastema cells (a highly proliferative tissue) [103]. Therefore, we propose that the autophagy mechanism involving the *ATG* gene family plays an important role in the formation of the elongated whip-like tail of *L. savala*. Based on the above analyses, we suggest that the *TES* gene family, *MTOR* gene, and *ATG* gene family play key regulatory roles in the formation of the specific body type *L. savala* (i.e., the elongated ribbon body axis, substantial number of vertebrae, and whip-like tail).

We also found that the *TRIO* gene family expanded significantly in the *L. savala* genome. *TRIO* encodes a large protein that functions as a GDP to GTP exchange factor, with a role in cell migration and growth by facilitating the reorganization of the actin cytoskeleton [104]. Chen et al. [105] confirmed that mice were born with shorter teeth and thinner dentin layers following the inactivation of *TRIO* in dental papilla mesenchymal cells. Further, in vitro cell culture assays showed that *TRIO* silencing resulted in the loss of proliferation and migration ability, and a higher apoptosis rate of human stem cells of the apical papilla (SCAPs) [105]. These results reveal that the *TRIO* gene acts as a positive mediator during the root formation and odontogenic differentiation of human SCAPs via the p38 signaling pathway [106]. Therefore, we speculate that the expansion of the *TRIO* gene family may drive the formation of sharp teeth in *L. savala*.

### 4.3. Movement and Immunity in L. savala

In positive selection analyses, genes involved in the immune-related pathways (e.g., complement and coagulation cascades) were significantly enriched, including *CFI*, *C1QA*, and *VTN*. The serine proteinase encoded by *CFI* plays a crucial role in the regulation of complement cascade reactions and the induced-fit factor responsible for controlling the complement-mediated processes [106]. It also participates in the regulation of the immune response [107]. *C1QA* encodes the A-chain polypeptide of serum complement subcomponent C1q [108]. C1q is a versatile innate immune molecule that combines with the proteases C1r and C1s to yield C1 [109], thus forming the first component of the serum complement system [110]. Complement proteins act synergistically to clear pathogens and induce a series of inflammatory responses to protect against infection and maintain immune homeostasis [111,112]. The complement system involving *CFI* and *C1QA* is an essential component of the innate immune response, and the first line of defence against pathogenic infections [113,114]. Vitronectin encoded by *VTN* is a cell adhesion and spreading factor in the serum and tissues [115]. A potential role of *VTN* was discovered in regulating the innate immunity of Japanese flounder *Paralichthys olivaceus* [116]. The *STAT6* gene and many interleukin-related genes (i.g., *IL5RA*, *IL10*, *IL15RA*, *IL16*, *IL17RA*, *IL20RA*, and *IL22RA2*) in the JAK-STAT signaling pathway and cytokine–cytokine receptor interaction pathway were screened in our positive selection analyses. *STAT6* encodes a member of the STAT family of transcription factors, with dual functions in signal transduction and transcriptional activation [117]. STAT6 contributes to defence against viral infection by mediating immune signaling in the endoplasmic reticulum [118]. As an important cytokine, interleukins encoded by the *IL* gene family play crucial roles in the intercellular signal transmission, activation, and regulation of immune cells [119]. In one of these, the protein encoded by *IL10* acts as an immunomodulatory cytokine, with pleiotropic effects in the immunoregulation and inflammatory response [120]. It could limit the excessive tissue disruption caused by inflammation [121].

Given that several genes under positive selection analyses were enriched in the immune-related pathways mentioned above, we can infer that *L. savala* has evolved a sophisticated immune system. This may be related to behavioural traits and motility in the species. *L. savala* is a predatory fish with better swimming ability and greater migratory behaviour than those of typical marine fishes [32,34,35]. Studies have demonstrated a strong correlation between immunity and exercise [122,123]. In juvenile Atlantic salmon *Salmo salar*, the inherent swimming performance and disease resistance have a positive correlation [124], and appropriate aerobic swimming exercises could promote growth and disease resistance [125]. Appropriate aerobic exercise improving antipredation and immunologic function was also revealed in the juvenile rock carp *Procypris rabaudi* [126]. In a study of water flow velocity focused on juvenile tinfoil barb *Barbonymus schwanenfeldii*, Zhu et al. [127] found that sustained aerobic swimming exercise improved the oxygen-carrying capacity and immune parameters. This indicated that swimming training could enhance the innate immune system of fishes [127]. Moreover, the domesticated and wild *S. salar* differ in swimming ability and immune responses, and the expression levels of immune-related genes (*CD40*, *C3-3*, *IL1B*, *CD276*, etc.) were significantly lower in the domesticated than in the wild *S. salar* with stronger swimming ability [125]. Therefore, we suggest that some immune-related genes undergo rapid evolution during the gain of aggressive predation and high-intensity swimming movements in *L. savala*, which may contribute to the sophisticated immune system.

In addition, we found that the *DNAH* gene family was significantly expanded in *L. savala*, and the enriched genes included *DNAH1* to *DNAH11*, except for *DNAH4*. The *DNAH* gene family encodes axonemal heavy chains associated with cell movement, which is involved in sperm flagellum assembly and motility [128,129]. Mutations in these genes could cause human sperm malformations [130,131]. Hu et al. [132] determined that sperm motility in Cyprinidae fishes is associated with high levels of gene expression in the *DNAH* gene family. Thus, we suggest that the expansion of the *DNAH* gene family may enhance the sperm motility of *L. savala*. However, studies of sperm motility in this species are lacking. Therefore, the specific regulatory relationship between sperm motility and the *DNAH* gene family in *L. savala* should be investigated further.

### 4.4. Contribution of DNA Repair Mechanisms to the Maintenance of Genomic Stability in L. savala

In our positive selection analyses, several genes associated with DNA repair were screened in the *L. savala* genome, including *POLM*, *PRKDC*, *BARD1*, *BRCA1*, *NBN*, *XRCC2*, *EME2*, and *FAAP100*. These genes were mainly enriched in three pathways, i.e., nonhomologous end-joining, homologous recombination, and Fanconi anemia. DNA polymerase Mu encoded by *POLM* participates in DNA double-strand break repair via the nonhomologous end-joining pathway [133,134]. There were five genes (*PRKDC*, *BARD1*, *BRCA1*, *NBN*, and *XRCC2*) involved in the homologous recombination pathway. During the gastrulation and early organogenesis of mice, the protein encoded by *PRKDC* promoted the repair of DNA double-strand breaks by combining with ATM (ataxia-telangiectasia mutated) to maintain its genomic stability [135]. The proteins encoded by *BARD1* and *BRCA1* combine to form a heterodimeric complex [136], which acts as a functional unit in mammalian cells in homologous recombination and DNA repair [137,138]. This heterodimer plays a role in DNA damage repair and transcriptional regulation to maintain genomic stability [139,140]. As a component of the MRN complex (MRE11-RAD50-NBN), the protein encoded by *NBN* is involved in DNA double-strand break repair and initiation of the DNA damage response to maintain genomic stability [141,142]. *XRCC2* encodes a member of the RecA/Rad51-related protein family, which participates in homologous recombination to maintain chromosome stability during cell division [143]. Based on cell cloning experiments of Chinese hamster *Cricetulus barabensis*, the chromosomal instability in cells with an *XRCC2* deficiency may be caused by defective homologous recombination [144]. Another two genes (*EME2* and *FAAP100*) were significantly enriched in the Fanconi anemia pathway. The protein encoded by *EME2* forms a heterodimer with MUS81 and functions as an XPF-type flap/fork endonuclease in DNA repair [145]. The protein encoded by *FAAP100* regulates FANCD2 monoubiquitination and the stability of the Fanconi anemia core complex, playing a role in the Fanconi anemia-associated DNA damage response [146]. In summary, the above-mentioned genes and pathways may play essential roles in the recombination of homologous chromosomes and maintenance of genomic stability in *L. savala*.

Additionally, we found that the *SLC6* gene family (*SLC6A1*, *SLC6A11*, *SLC6A13*, and *SLC6A19*) and the *COL4* gene family (*COL4A1*, *COL4A2*, and *COL4A6*) were significantly contracted in the *L. savala* genome. *SLC6* is involved in the transport of neurotransmitters (e.g., dopamine, norepinephrine, serotonin, GABA, and glycine) [147,148]. This suggests that the contraction of the *SLC6* gene family may be related to nervous system evolution in *L. savala*. Type IV collagen encoded by the *COL4* gene family is the major component of the basement membrane in many tissues [149]. It plays a pivotal role in the remodeling of endometrial tissue in mammals by regulating the structure, viability, and differentiation of endometrial cells [150,151]. On this basis, we speculate that the *COL4* gene family is related to the formation of the smooth body surface in *L. savala*. Owing to the lack of detailed information on the nervous system and body surface development in *L. savala*, it is difficult to clearly establish the effects of the contractions of these two gene families on these traits, and further studies are needed.

## 5. Conclusions

In this study, we obtained a high-quality chromosomal-level genome assembly of *L. savala*, providing the first genomic dataset for trichiurid fishes. Based on comparative genomic analyses, we found that *MTOR* gene, and the *TES*, *ATG*, and *TRIO* gene families may be key factors driving the formation of the unique body shape and sharp teeth in *L. savala*. Moreover, several immune-related genes (*CFI*, *C1QA*, *VTN*, *STAT6* genes, and the *IL* gene family) underwent rapid evolution, likely contributing to the sophisticated immune system in *L. savala*. These changes may also be related to the evolution of aggressive predation and intense swimming movements in this species. In addition, DNA repair mechanisms may play crucial roles in maintaining the evolutionary stability of the *L. savala* genome. Our study preliminarily reveals the molecular mechanisms underlying the special morphological and behavioural characteristics of *L. savala* at the genomic level, and provides an invaluable reference for genomic and evolutionary studies of other trichiurid fishes.

## Figures and Tables

**Figure 1 genes-14-01268-f001:**
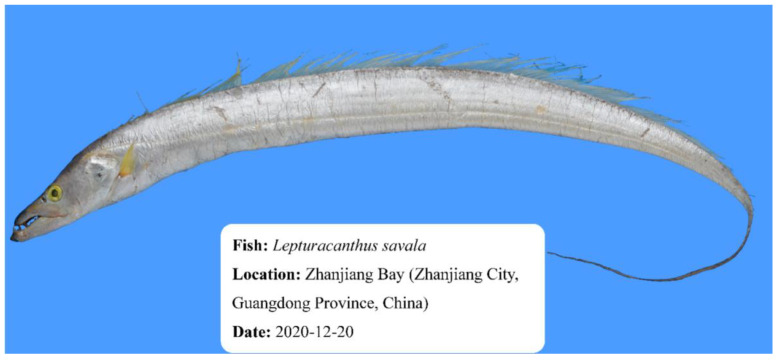
High-definition image, sampling location, and sampling date of *L. savala* used for genome sequencing.

**Figure 2 genes-14-01268-f002:**
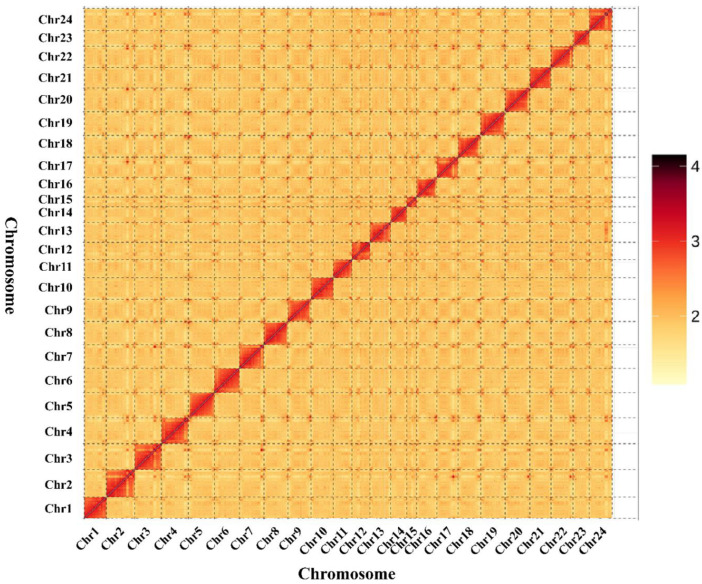
Statistics of the Hi-C assembly of the *L. savala* genome. The colour reflects the intensity of each contact, with deeper colours representing higher intensity.

**Figure 3 genes-14-01268-f003:**
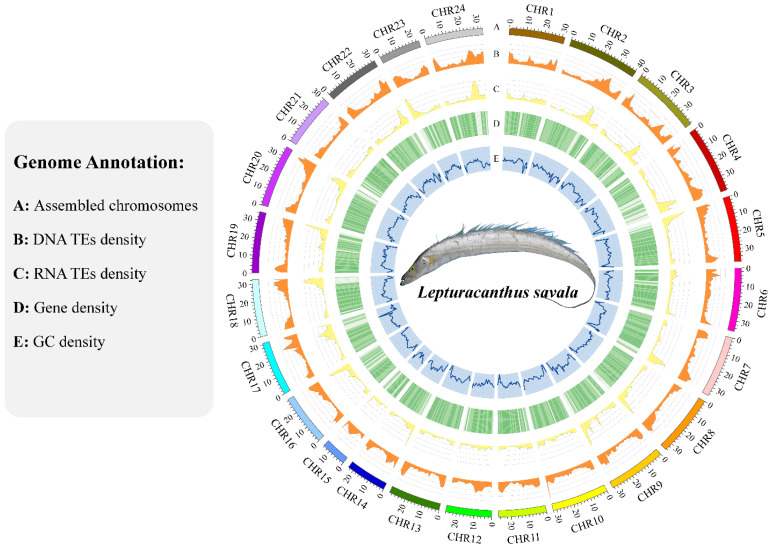
Genome coordinates and annotation information of the *L. savala* genome.

**Figure 4 genes-14-01268-f004:**
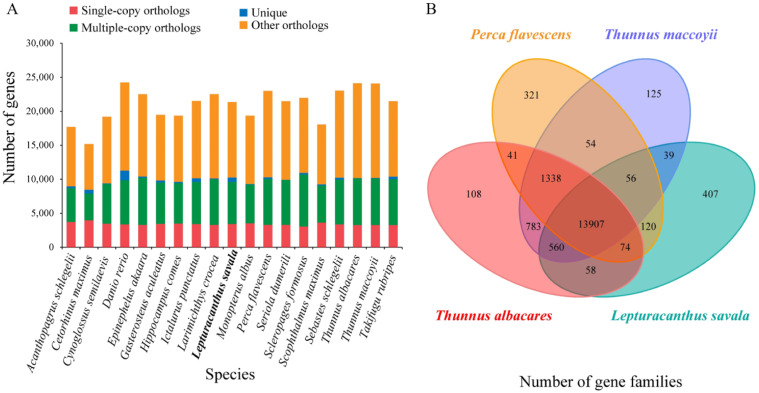
Types and numbers of gene families in 19 species (**A**) and quantitative analysis of gene families in *L. savala*, *T. albacares*, *T. maccoyii*, and *P. flavescens* (**B**).

**Figure 5 genes-14-01268-f005:**
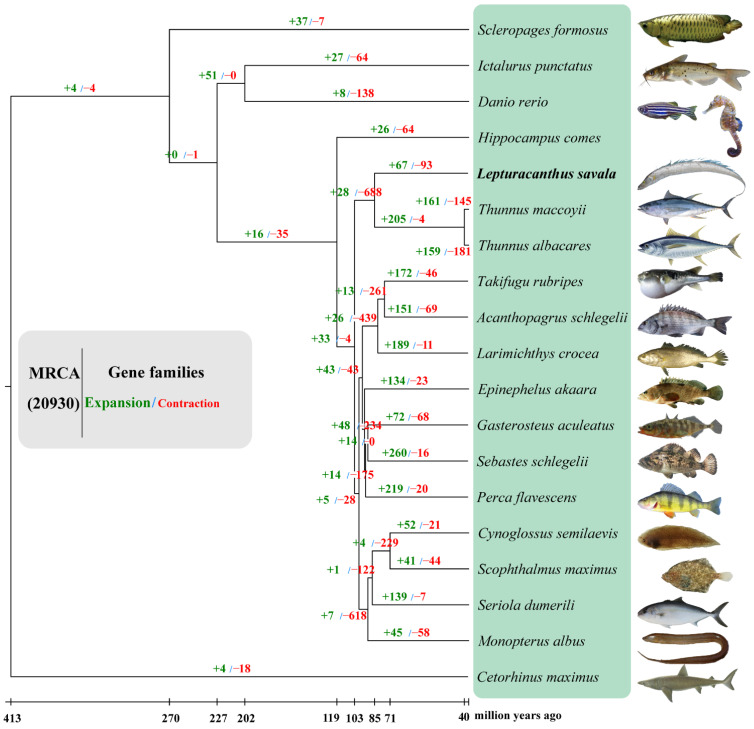
Gene family expansions and contractions for 19 species.

**Figure 6 genes-14-01268-f006:**
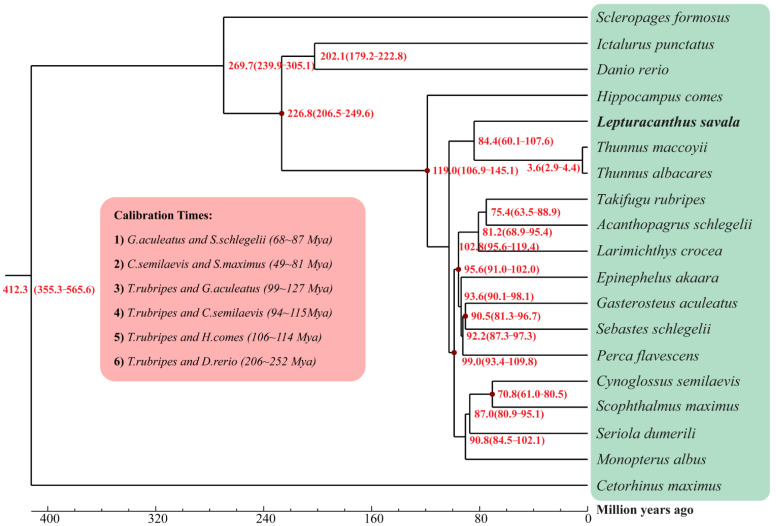
Evolutionary divergence times of 19 species.

**Table 1 genes-14-01268-t001:** Structure and parameters of genes predicted by three different methods.

Methods	Gene Set	Number	Average Transcript Length (bp)	Average CDS Length (bp)	Average Exons per Gene	Average Exon Length (bp)	Average Intron Length (bp)
De novo	Augustus	33,486	8827.52	1201.39	6.87	174.85	1298.98
	GlimmerHMM	76,383	8606.22	686.71	4.6	149.37	2201.43
	SNAP	64,956	11,484.45	799.65	5.76	138.71	2242.37
	Geneid	34,350	14,311.26	1231.61	6.09	202.2	2569.11
	GenScan	33,084	16,121.64	1490.15	8.22	181.34	2027.19
Homolog	*Danio rerio*	23,256	10,252.69	1494.80	7.53	198.53	1341.28
	*Etheostoma spectabile*	22,973	11,643.93	1604.77	8.3	193.26	1374.52
	*Gasterosteus aculeatus*	27,165	8780.31	1235.59	6.74	183.24	1313.76
	*Homo sapiens*	18,148	11,303.80	1465.64	7.95	184.38	1415.76
	*Oryzias latipes*	23,446	11,206.10	1630.43	8.19	199.16	1332.48
	*Perca flavescens*	26,554	10,560.30	1481.67	7.73	191.65	1348.74
	*Perca fluviatilis*	25,661	10,889.74	1529.66	7.87	194.37	1362.46
	*Sander lucioperca*	25,313	11,255.57	1578.82	8.21	192.27	1341.86
	*Thunnus albacares*	25,361	11,411.78	1589.75	8.25	192.62	1354.14
	*Thunnus maccoyii*	24,446	11,716.77	1631.43	8.48	192.5	1349.21
	*Takifugu rubripes*	22,038	11,995.55	1634.46	8.63	189.42	1358.19
RNA-Seq	PASA	43,445	11,533.68	1469.51	9.05	162.38	1250.24
	Cufflinks	37,916	13,980.13	2755.81	8.75	315.05	1448.83
EVM (EVidenceModeler)		31,876	10,753.39	1307.47	7.49	174.54	1455.28
PASA-update *		31,434	11,153.72	1339.72	7.67	174.73	1471.95
Final set **		23,625	13,717.34	1620.82	9.38	172.74	1442.96

*: Contains UTR region. **: This final set contains UTR region.

**Table 2 genes-14-01268-t002:** KEGG enrichment analysis of expanded and contracted gene families.

**1. Contraction (93 Gene Families, 13 KEGG Pathways)**		
KEGG pathways	*p*-value	Genes
Synaptic vesicle cycle	1.61 × 10^−6^	*SLC6A13*, *SLC6A1*, *SLC6A11*
GABAergic synapse	5.39 × 10^−6^	*SLC6A13*, *SLC6A1*, *SLC6A11*
Choline metabolism in cancer	7.27 × 10^−5^	*SLC22A5*, *SLC5A7*
NOD-like receptor signaling pathway	0.0028841	*NLRC3*, *GVIN1*, *URGCP*
Small cell lung cancer	0.0040207	*COL4A1*, *COL4A2*, *COL4A6*
Protein digestion and absorption	0.0044306	*COL4A1*, *COL4A2*, *COL6A3*, *SLC6A19*, *COL4A6*, *SLC6A19*
Pathogenic Escherichia coli infection	0.0060438	*TUBB1*, *COL6A3*
Necroptosis	0.0111952	*NLRC3*, *COL6A3*, *CAPN2*, *ALOX5*
Mineral absorption	0.0208705	*SLC6A19*
Gap junction	0.0305233	*TUBB1*, *COL6A3*
Arachidonic acid metabolism	0.0311748	*ALOX5*
ECM–receptor interaction	0.0359224	*COL4A1*, *COL4A2*, *COL6A3*, *COL6A6*
Focal adhesion	0.0466035	*COL4A1*, *COL4A2*, *COL6A3*, *COL6A6*
**2. Expansion (67 gene families, 18 KEGG pathways)**		
KEGG pathways	*p*-value	Genes
Focal adhesion	0.00	*TRIO*, *TES*
ECM–receptor interaction	0.00	*TRIO*, *TES*
Platelet activation	0.00	*TRIO*, *TES*
Relaxin signaling pathway	0.00	*TRIO*, *TES*
AGE-RAGE signaling pathway in diabetic complications	0.00	*TRIO*, *TES*
Protein digestion and absorption	0.00	*TRIO*, *TES*
Amoebiasis	0.00	*TRIO*, *IGHM*, *GPR119*, *TES*
Human papillomavirus infection	1.36 × 10^−262^	*TRIO*, *F5*, *EIF3A*, *TES*
PI3K-Akt signaling pathway	6.51 × 10^−251^	*TRIO*, *IGHM*, *TES*
Olfactory transduction	4.99 × 10^−40^	*NONE*
Lysine degradation	2.09 × 10^−6^	*KMT5AA*, *KMT5A*, *SET-1*
Huntington disease	4.89 × 10^−5^	*DNAH7*, *DNAH11*, *DNAH9*, *NES*, *DNAH3*, *DNAH5*, *DNAH8*, *DNAH2*, *DHC10*, *KLF18*, *SGS4*, *DNAH1*, *DNAH6*, *QRICH2*, *DNAH10*
Staphylococcus aureus infection	0.0079023	*IGLV1-51*, *IGHM*, *SFTPD*, *MBL*, *MBL2*, *IFITM3*
Cortisol synthesis and secretion	0.0119558	*CACNA1G*, *CACNA1H*, *CACNA1I*, *CACNA1H*
PPAR signaling pathway	0.0169356	*SAMD3*
Bacterial secretion system	0.017091	*SECA3*, *SECA*
Allograft rejection	0.0267611	*IGLV1-51*, *IGHM*, *PRF1*
Glycosphingolipid biosynthesis	0.0462553	*ST3GAL1*

**Table 3 genes-14-01268-t003:** Genome characteristics comparison based on two groups of positive selection analyses.

**Group 1 (Genes: 903; GO Terms: 62; KEGG Pathways: 17)**		
A: *L. savala*; B. *A. schlegelii*, *L. crocea*, *P. flavescens*		
GO terms	KEGG Pathways	Genes screened
DNA metabolic process	JAK-STAT signaling pathway	*HIRA*, *IL15RA*, *PRLR*, etc.
Nuclear chromosome	Novobiocin biosynthesis	*TAT*
DNA repair	Fanconi anemia pathway	*EME2*, *FAAP100*, *BRCA1*, etc.
Nucleic acid binding	Cytokine–cytokine receptor interaction	*INHBA*, *HIRA*, *TNFRSF26*, etc.
Helicase activity	Sulfur relay system	*SYNPR*, *MOCS2*, *NFS1*
Nuclease activity	Ether lipid metabolism	*TPT1*, *SH3BGRL3*, *PLA2G3*, etc.
Checkpoint clamp complex	Arginine biosynthesis	*NOS1*, *ASL*, *NAGS*, *GLS2*
Spindle	RNA transport	*RANBP2*, *EIF2B3*, *RPP30*, etc.
Hyaluronic acid binding	Homologous recombination	*BARD1*, *EME2*, *BRCA1*, etc.
Chromatin binding	Phenylalanine, tyrosine, and tryptophan biosynthesis	*TAT*
Ino80 complex	Tropane, piperidine, and pyridine alkaloid biosynthesis	*TAT*
Protein homodimerization activity	Alanine, aspartate, and glutamate metabolism	*ASNS*, *ASL*, *ABAT*, etc.
7S RNA binding	Ubiquinone and other terpenoid-quinone biosynthesis	*TAT*, *COQ6*
Signal recognition particle	Thiamine metabolism	*AK5*, *CFAP61*, *NFS1*
ATPase activity	Ribosome biogenesis in eukaryotes	*UTP14A*, *RIOK1*, *HEATR1*, etc.
Isomerase activity	Nonhomologous end-joining	*PRKDC*, *POLM*
DNA damage checkpoint	Complement and coagulation cascades	*F5*, *PLAU*, *CPB2*, etc.
**Group 2 (Genes: 922; GO terms: 70; KEGG Pathways: 18)**		
A. *L. savala*, *M. albus*; B. *A. schlegelii*, *L. crocea*, *P. flavescens*		
GO terms	KEGG Pathways	Genes screened
Methyltransferase activity	Cytokine–cytokine receptor interaction	*TNFRSF13B*, *OSMR*, *PRLR*, etc.
Aminomethyltransferase activity	JAK-STAT signaling pathway	*OSMR*, *PRLR*, *IL15RA*, etc.
Nucleic acid binding	RNA transport	*UPF3A*, *EIF3F*, *EIF3C*, etc.
Neurotransmitter metabolic process	Thyroid cancer	*ANKDD1A*, *RET*, *CCDC6*, etc.
Catabolic process	Ribosome biogenesis in eukaryotes	*HEATR1*, *REXO1*, *VSTM2A*, etc.
Organic substance catabolic process	Autophagy—other	*ATG3*, *TRIM14*, *MTOR*, etc.
Glycine catabolic process	Pancreatic cancer	*E2F3*, *ANKDD1A*, *VEGFAA*, etc.
RNA cap binding complex	Intestinal immune network for IgA production	*TNFRSF13B*, *IL15RA*, *CD28*, etc.
RNA binding	Autophagy—yeast	*ATG3*, *TRIM14*, *MTOR*, etc.
Organonitrogen compound catabolic process	Chronic myeloid leukemia	*E2F3*, *ANKDD1A*, *GRAP*, etc.
LUBAC complex	EGFR tyrosine kinase inhibitor resistance	*VEGFAA*, *MTOR*, *GRAP*, etc.
N-methyltransferase activity	Fanconi anemia pathway	*BRCA1*, *ANKDD1A*, *RMI1*, etc.
Phospholipase A2 activity	Phenazine biosynthesis	*PBLD*
Drug catabolic process	Ubiquinone and other terpenoid-quinone biosynthesis	*COQ2*, *TAT*, *COQ6*
Threonine-type endopeptidase activity	Glycine, serine, and threonine metabolism	*AMT*, *CHDH*, *DMGDH*, etc.
Proteasome core complex	Nonhomologous end-joining	*XRCC6*, *DCLRE1C*, *DNTT*
Kinetochore	Prostate cancer	*E2F3*, *MTOR*, *GRAP*, *BAD*, etc.
Protein homodimerization activity	Acute myeloid leukemia	*MTOR*, *GRAP*, *BAD*, etc.

## Data Availability

The genome assembly data of *Lepturacanthus savala* was deposited at NCBI under BioProject number PRJNA953192 (Submitted), BioSample number SAMN34109439 (Submitted). The raw read sequence accession numbers: SRR24890858, SRR24890859, SRR24890860, SRR24890861, SRR24890862, SRR24890863, SRR24890864.

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
