# Peer review of "Chromosome-Level Genome Assembly Provides Insights into the Evolution of the Special Morphology and Behaviour of Lepturacanthus savala"

_genes, 2023, doi:10.3390/genes14061268_

Round 1

Reviewer 1 Report

To Authors,

Thank you so much for the nice and diligent work. Savalani hairtail is a very well-known fishery species in China. It carries a lot of good memories of tasty family meals and happy holiday times for almost all Chinese born during the 1960s to 1990s. As a marine fish biologist, I was amazed that no genomic-scale work had previously been done for this species. The current study is definitely a long-overdue work that our marine biologist owes to the Chinese people and is worth publishing.

The study combined a lot of good-quality sequencing results and presented us with the first chromosome-level genome assembly for the species. On top of the phylogenetic analysis and species divergence time estimation, the comparative genomic analysis revealed some important potential mechanisms underlying hairtail's unique morphology and behaviour characteristics.

However, I would not suggest the manuscript be published in its current form. I am fully aware of the difficulties of not being a native English language speaker, but still, the proper academic language should be present in the manuscript. Quite a few extremely long sentences in the text contain mixed messages authors want to express but created a lot of possible confusion and misunderstandings. I would suggest the authors use simple structure sentences and just try to deliver one message at a time. The editor informed me that they have an in-house English editing department that would help you with the language editing. I’ll leave you to their good hands then.

Except for the language use, another thing I what to address is that not all the valuable results were sufficiently discussed. For example, the phylogenetic analysis results, does results confirm or deny the previous phylogenetic studies. Does the divergence time of hairtail with other closely related species correspond to any of the geology history events? Possible species origins? Please consider adding a few more phylogenetic literatures in the introduction so that you can carry on the phylogenetic analysis result discussion in the latter part of the manuscript.

The references need to be fixed too. For example, some of the conference proceedings and books are missing publishers and places to publish. And for #30 Yi M. (2019). Based on skeletal comparison and COI sequence analysis for 6 species of cutlassfishes Trichiuridae systematic classification in Chinese coastal water. Guangdong Ocean Univ. missing page number 1–4. and doi number: doi: 10.27788/d.cnki.ggdhy.2019.000065

Finally, please consider deleting the NCBI email corresponding from the supplementary files. Just simply providing the accession number in the text would be good enough.

I would not suggest the manuscript be published in its current form. I am fully aware of the difficulties of not being a native English language speaker, but still, the proper academic language should be present in the manuscript. Quite a few extremely long sentences in the text contain mixed messages authors want to express but created a lot of possible confusion and misunderstandings. I would suggest the authors use simple structure sentences and just try to deliver one message at a time. The editor informed me that they have an in-house English editing department that would help you with the language editing. I’ll leave you to their good hands then.

Reviewer 2 Report

General comment:

In this paper, the authors generate for the first time a chromosome-level assembly of the genome of L. savala, providing information about the functional annotation, as well as phylogenetic analysis of the species. Additionally, the authors showed an evolutionary analysis of gene families in this species. The manuscript is appropriate and generates novel and useful knowledge about this fish species. However, some corrections are required, mainly focusing on more methodological details that should be provided.

Specific comments:

-There are some minor redaction problems to solve in the manuscript.

-The common name should also be included in the abstract. Additionally, the abstract extension should be a maximum of 200 words according to journal instructions for authors.

Introduction:

-The second paragraph of the introduction should be revisited because the present way is not clear and is redacted in a confusing way.

-In line 107, the sentence “The life cycle 107 of L. savala involves greater degrees of movement and motility than those of other ordinary marine fishes” should be supported by the respective reference.

Materials and Methods:

-In section 2.1, the bioethical committee approval certificate for this study should be presented in the text (the approbation number), considering the manipulation of live fish. Additionally, the euthanasia method, anesthetic used, and dose should be provided.

-In section 2.2, the purification protocol of the DNA should be included. Additionally, the library kits for Illumina sequencing should be included, as well as the version of the library’s kits. DNA and RNA quantification and integrity evaluation should be included. Integrity is crucial for successful sequencing, therefor quality measures should be included also in the methodology (ej. RQN or RIN).

-In point 2.3, more details should be provided about genome size estimation.

-In point 2.4, a more detailed methodology should be incorporated in terms of parameters of the assembly, as well as SNPs annotation.

-More details in the parameters used for annotation should be included.

Results:

-In line 286, more support should be incorporated for this affirmation.

-The legend of figure 2 and the figure should indicate unit measure (HiC contact count).

-The raw read sequences accession number should also be included.

-In figure 4, the A and B letters should appear in the figure.

-Resolution of figure 5 and 6 should be improved.
